# The influence mechanism of work–family conflict on employees' proactive behavior: A moderated mediator model

Hongyuan Zhang[1,2] Yule Wan[2] Shuming Zhao[3]

1 Institute of Marine Economics, Jiangsu Ocean University, Lianyungang, China, 2 Business school of Jiangsu Ocean University, Lianyungang, China, 3 School of Business, Nanjing University, Nanjing, China

## Abstract

For employees, work and family are two important facets of their life development. Work–family conflict has a significant impact on employees' proactive behavior. This study used questionnaires to gather information from 126 companies, 141 teams, and 1354 employees based on the resource conservation theory. The study discovered that employees' proactive behavior is negatively impacted by work–family conflict, with emotional exhaustion acting as a mediating factor. The association between employees' proactive behavior and emotional exhaustion is negatively moderated by the high commitment work system. The effect of emotional exhaustion on employees' proactive behavior is less in organizations that have adopted a high commitment work system than in those that have not, or that have low levels of commitment in their human resource management practices. Enhancing the current studies and advancing our understanding of the relationship between work–family conflict and employee proactive behavior, this paper enhances the body of knowledge on the subject. Additionally, it offers companies practical work–family conflict resolution techniques that can improve employee efficiency and assist the company in reaching its objectives.

## 1 Introduction

High–quality development is the main goal for fully constructing a modern socialist nation, according to the report to the 20th CPC National Congress [1]. Companies have been crucial in fostering high–quality growth, and they are now the center of attention for all facets of society as they work to strengthen their core competencies and increase their competitiveness. The enterprise's external environment is complicated and dynamic right now, and competition is intense. In this instance, the collective efforts of the employees are inextricably linked to the growth of the company at large. The more proactive employees are, the more efficient and productive they will be. At the same time, it can reduce the management cost of enterprises, improve the operation efficiency of organizations, and enhance the competitiveness of enterprises [2]. As communication technology has advanced, remote communication has become increasingly convenient. Employees' work environments also affect their families, which makes them feel bad and, thus, leads to bad work habits [3]. As a result, the research of employees' proactive behavior has taken center stage.

**Data availability statement:** The data has been uploaded to a public database and can be viewed at the following link: https://figshare.com/s/16cd5be591b5c22560f7

**Funding:** The article was supported by certain funding from two funding projects, mainly for questionnaire surveys. In addition, the sponsors helped coordinate with some enterprises for interviews and distributing questionnaires during the author's research: National Natural Science Foundation of China Special Project: "Research on Digital Human Resource Development and Management within Human-Machine Interaction Scenarios" (project number 7234-2027 to S.Z.); The 6th Phase of the "521 Project" for Science and Technology Funding in Lianyungang City: "Research on the Pathways for Lianyungang to Expeditiously Build a Prominent Modern Ocean Industrial System" (project number LYG0 6521 202391 to H.Z.). Research on the Relationship between Internationalization of Human Resources and International Entrepreneurial Performance from the Perspective of Endogenous Growth (project number KYCX2023-117 to Y.W.).

**Competing interests:** The authors have declared that no competing interests exist.

At present, most of the research on the antecedents of employees' proactive behavior focuses on factors unique to the workplace, such as leadership style and job attributes [4–6]. Although scholars have previously discussed the influencing elements in non–work domains, the lines separating work and non–work regions are becoming increasingly hazy due to the rapid advancement of mobile internet technology. The relationships between these two areas have been overlooked in previous studies. For employees, work and family are two important factors [7]. As the most important non–work area for employees, the family is intimately related to, dependent upon, and influences work. It is impossible to overlook the impact of employees' work–family relationships on them [8]. It is important to examine how work-family relationships affect employees' proactive behavior. Population aging is a major issue in China, where fertility is promoted, the three–child policy is in place, the percentage of dual–income families is rising, issues with caring for the elderly, supporting children and family, and work–related issues like overtime and telecommuting are piling up. Employees must devote a limited amount of their energy to work and family, and work–family conflict is becoming a bigger issue [9]. Work–family issues will undoubtedly significantly influence employees' proactive conduct when combined with the current national circumstances in China. According to resource conservation theory, people are more willing to protect and acquire resources that they consider important, and they will view possible or actual resource loss as a risk [10]. Work and family are interwoven for employees with distinct duties in both. When employees perform numerous responsibilities concurrently, a lack of resources will result in behavior disorders, emotional exhaustion [11]. Employee initiative and efficiency are affected when their resources are compromised at work, leading to increased stress and a decline in job satisfaction [12]. Thus, employees' negative emotions may play a role in this situation [13]. Previous studies have shown that the human resource management mode of enterprises impacts employees' behaviors [14,15]. Therefore, this paper believes that work-family conflict, enterprise organization or leader's human resource management style may affect employees' proactive behavior.

In conclusion, this paper aims to investigate practical strategies for improving proactive behavior among employees in China's current workplace. It explores how work–family conflict affects employees' proactive behavior by looking at the problem from the standpoint of work–family conflict and using resource conservation theory as a foundation. It also looks at the moderating influence of high commitment work systems and the mediating function of emotional exhaustion. This paper advances our knowledge of the connection between work–family conflict and employees' proactive behavior, offering businesses–focused human resource management tactics to boost employee efficiency and eventually accomplish corporate objectives.

## 2 Theoretical basis and research hypothesis

### 2.1 Work–family conflict and employees' proactive behavior

The incompatibility of work and family needs leads to an inter–role conflict known as work–family conflict [16]. There are two main types of work–family conflict:

work–family conflict, which arises when meeting work needs gets in the way of family obligations; family–work conflict, which arises when meeting family needs gets in the way of regular work activities [17]. Employees typically prioritize their work [18], and their proactive behavior is a sort of spontaneous behavior to change the environment and optimize oneself [19]. This is because Chinese organizations, which are influenced by a collectivist culture, encourage employees to sacrifice their families for the benefit of the group. The term "proactive behavior" describes a range of intentional or unplanned actions staff members take to finish duties at work effectively. Employees will impulsively engage in various actions to guarantee the successful accomplishment of work goals; these actions are proactive and characterized by zeal and persistence [20]. According to the resource conservation theory, people with work–family conflicts tend to exhibit fewer altruistic and positive behaviors to make up for or prevent the consumption of internal resources, which are continuously depleted, negative emotions and behavioral disorders are also more likely to occur [21]. According to empirical research, Xu Yan (2024) discovered that work–family conflict has a significant negative impact on employees' work performance [22], Chinese scholar Li Yifei et al. (2019) found through empirical analysis that work–family conflict hurts job satisfaction [23], and Zhang Junwei et al. (2023)discovered that work–family conflict can significantly affect performance as goals by the resource conservation theory [24]. As a result, work–family conflict can decrease employees' proactive behavior. Based on these findings, the article puts forth the following hypothesis:

Hypothesis 1: Work–family conflict reduce employees' proactive behavior.

## 2.2 The mediating effect of emotional exhaustion on employees' proactive behavior

Emotional exhaustion is a psychological state, which belongs to the core dimension of the variable job burnout, and is the physical and emotional depletion caused by various stresses [25]. Employees who experience work–family conflict may become emotionally exhausted [26]. The idea of resource conservation uses the perspective of resource gain and loss to explain how emotional weariness is produced. For employees, positive emotions are a very important resource. When emotional resources are continuously depleted to a certain extent, it will reduce their self-restraint ability, making it impossible for them to control their own behavior and meet the requirements of their roles [27]. Employees who suffer work–family conflict will be physically and emotionally absorbed at work on one level, and their resources will be overcrowded on the other [28]. Emotional exhaustion will occur from employees' attempts to control these negative emotions, which will further drain their emotional reserves. Previous studies show that employees' proactive behaviors are significantly affected by work–family conflict [29]. Additionally, the emotional of employees have a direct impact on their behaviors [30]. Proactive hebaviors necessitates a certain amount of mental, emotional, and physical energy from employees [31]. Theory and practice have proved that employees tend to help colleagues and maintain the organization when they are in a happy mood. In other words, positive psychological resources help to promote employees' proactive behaviors; Conversely, when the positive psychological resources are reduced, the proactive behavior of employees will also decrease [32].

The resource conservation theory states that when people experience a loss of resources, they typically refrain from engaging in harmful activities because they are under psychological stress [33]. A person experiencing negative emotional reactions tends to display reduced enthusiasm and a poorer attitude towards work, ultimately leading to burnout and diminished commitment to the company, potentially initiating a trend towards turnover [34]. In order to minimize the loss, employees typically cut back on their resource input right away, which frequently results in a gradual decline in their proactive behaviors. Employees may thus suffer from emotional exhaustion as a result of work–family conflict, which can result in an unequal distribution of resources between the family and the workplace. Employees experience tension when their emotional reserves are not sufficiently exhausted. The resources conservation theory predicts employees in this condition will be less proactive in preventing more resource loss. Furthermore, a number of empirical research have shown that the relationship between work–family conflict and proactive conduct on the part of employees is mediated by emotional weariness [32,35]. In light of these evidence, we put out the following hypothesis:

Hypothesis 2: Emotional exhaustion plays a mediating role in the negative impact of work–family conflict on employees' proactive behavior.

## 2.3 The moderating role of high commitment work system

The term "high commitment work system" refers to a set of strategic human resource management practices intended to increase employees' commitment to the company [36]. The system's management practices include careful hiring and selection, consideration of the degree of alignment between the quality of employees and job positions and the needs of the company, and the use of various incentive programs and development opportunities to encourage employees to adopt work behaviors that align with the company's expectations and needs. Additionally, the system strengthens the alignment between employees and positions and between employees and corporate values [37]. Work–family conflict will use up a variety of people's resources. In the human resource management practice of putting in place a high commitment work system, some external conditions can be provided for employees to obtain external resources [38]. As a result, companies that adopt a high commitment work system can effectively reduce the negative effects of work–family conflict on employees. Min (2022) stated that a high–commitment work system encourages initiative and creativity while reducing negative emotions [39]. As a result, the impact of changes in individual emotions, capital, and other resources on workers' active behaviors in the high commitment work system will be part of the institutional boundary effect.

Companies will strengthen knowledge and skill training, create a varied and long–term performance evaluation system to enhance employees' self–development behaviors and promotion channels in the internal labor market [40], and set up a flexible and equitable internal and external salary system as part of the high commitment work system's human resource management practices. Implementing employee involvement plans and flexible work designs serves to foster diverse communication and solicit employee input [40]. Within this framework, employees are regarded as integral partners in a high commitment work system. Through active communication and various channels, the company can gain a profound understanding of employees' requirements and effectively convey corporate goals. This, in turn, fosters acknowledgment among workers and, through emotional commitment, facilitates the achievement of shared objectives and values between the company and its employees. According to earlier research, companies that apply a high commitment work system can increase employee motivation [41], sense of justice [42], trust, and organizational support [43], as well as increase job satisfaction and lower stress levels [44]. Employees in high commitment work systems are more likely to value their positions in both work and life or receive support and assistance from the company. Employees may feel somewhat psychologically pleased and less emotionally exhausted as a result of the organization's understanding and support. Therefore, the adoption of the high commitment work system allows employees to acquire new resources that they value in the business to compensate for the loss of their resources when they feel emotional weariness [40]. Therefore, through a variety of psychological processes, employees in a high commitment work system can consciously transform positive emotions, gaining access to internal emotional resources and reducing the impact of emotional exhaustion on proactive behaviors. In light of this, the following hypotheses are put forth:

Hypothesis 3: The impact of emotional exhaustion on proactive conduct by employees is negatively moderated by a high commitment work structure.

In addition to the aforementioned hypotheses, we propose that the degree of high commitment work systems that firms have in place may also have an impact on how work–family conflict affects employees' active behavior through emotional weariness. The social exchange hypothesis states that if one agent benefits another, the other would freely repay the favor by paying the other person something in return [45]. Based on the costs and advantages of the trade contact, the individuals participating in the relationship will assess the quality of the relationship. In an organization, people will freely contribute to it if they accept the benefits that the company offers. According to the reciprocity principle, when an employer shows concern and care for their employees, the employees will return the favor by acting and thinking more positively [46]. The establishment of a high commitment work system can furnish employees with essential human capital, emotional

resources, and social capital, thereby promoting productive behaviors that contribute positively to organizational outcomes. Under the high commitment work system, the organization not only enhances employees' skills but also communicates to them the employer's expectation that they would adopt proactive behaviors by assessing their current skill level and providing focused training [47]. Jongwook (2023) noted that workers will have a strong feeling of duty to return as the high commitment work system offers competitive pay and job stability. They are therefore more inclined to take creative chances [48]. In light of this, the following hypothesis is put forth:

Hypothesis 4: Through emotional exhaustion, the high commitment work system mitigates the indirect impact of work–family conflict on employees' proactive behavior.

## 2.4 Theoretical model

This study examines how work–family conflict affects employees' proactive behavior from the viewpoints of resource conservation theory by reviewing and evaluating pertinent literature on employee initiative behavior, work–family conflict, emotional exhaustion, and high commitment work system. It introduces emotional exhaustion as the mediating variable and high commitment work system as the moderating variable. The image below displays the paper's research hypothesis model (Fig 1):

# 3 Research design

## 3.1 Research methodology

Data for this paper was gathered offline through the distribution of paper questionnaires. MPLUS can directly implement the syntax instructions for the analysis of moderated effects of latent variables, obtain the regression coefficients and corresponding significance of the independent and dependent variables under low and high moderation variables, and draw pictures—all of which are more appropriate for this study because it must examine the mediating role of emotional exhaustion throughout the process and the moderating role of a high commitment work system. Consequently, SPSS 25.0 and MPLUS statistical tools were used to process and evaluate the data in order to validate the hypothesis. This study, which involves human subjects, has been approved by Jiangsu Ocean University's Academic Committee for ethical assessment. The ethical review board approved the application for the survey on 8/5/2024, with the survey commencing on 15/5/2024 and concluding on 24/8/2024. Written informed consent was obtained from all respondents, and consent was also obtained for the follow–up survey. It is noteworthy that all questionnaires were completed anonymously, thereby ensuring the privacy of the respondents was not compromised. The original ethical review board approval and informed consent forms were uploaded into the system as supporting documentation. Since this survey is for employees of enterprises, it is illegal for enterprises to hire employees under the age of 18 in China, so minors (less than 18 years old) are excluded from this survey. Written informed consent forms were distributed to all participants, and only after obtaining their consent was the subsequent survey carried out. All questionnaires were filled out anonymously to ensure no exposure to the respondents' privacy.

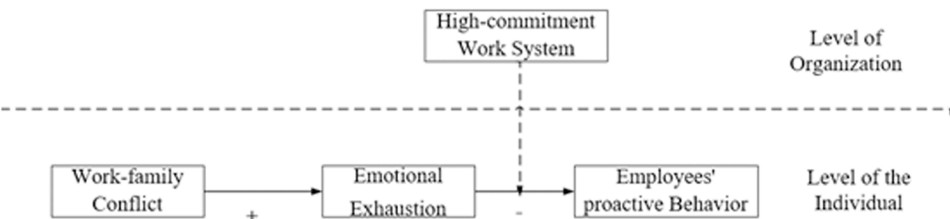

**Fig 1. Research hypothesis model.**

## 3.2 Data collection

This paper conducted a small sample pre–survey on a few enterprises prior to the official distribution of questionnaires in order to test the efficacy of the measurement scale used in the paper and the rationality of the main effect of the research model built in the paper. A formal survey was carried out following the preliminary survey's verification of the primary effect's reasonableness.

**3.2.1 Pre–research stage.** Data collection is completed through on–site distribution and recovery of paper questionnaires. In this stage, the questionnaire was reviewed and screened, mainly by referring to the screening methods of most scholars, that is, to check whether there were more information gaps in the questionnaire, whether the answers to multiple consecutive questions in the questionnaire were the same or the answers were filled in with strong regularity. In the above two cases, the questionnaire was regarded as invalid and should be eliminated. In the pre–research stage, a total of 90 questionnaires were sent out, and 78 were effectively collected, with an effective recovery rate of 86.7%.

In the pre–research stage, there were 32 females, accounting for 41.03% of the total sample. Those aged between 26 and 30 accounted for 26.92%, and 78.21% of the respondents had a bachelor's degree or above, indicating that the majority of respondents to the test were highly educated people. 83.33% of the employees have more than 3 years of work experience in the workplace at the time of the survey. Among the respondents, 27 (34.62% of the total) are from the field of sales, and 29 (37.18% of the total) are from the field of functional management. The remaining 22 (28.21% of the total) were from other professional fields such as production. In addition, the marital status survey showed that respondents are sensitive to personal information such as divorce and widowhood, and only two options will be reserved for the formal survey: single and married.

In the pre–survey questionnaire, measures of work–family conflict, emotional exhaustion, and employee's active behavior are listed, among which the reliability of work–family conflict is 0.902, that of emotional exhaustion is 0.886, and that of employee's active behavior is 0.907. The reliability of relevant scales meets the requirements. In addition, through the analysis of preliminary test results, it can be verified that there is a certain correlation between work–family conflict and employees' active behavior.

**3.2.2 Formal investigation stage.** This paper has issued questionnaires to 126 enterprises, covering industries including manufacturing, Internet, medicine, new energy, and so on. In this paper, 1,561 employee questionnaires were distributed and 1,412 were recovered, among which 1,354 were valid and 58 were invalid. The effective recovery rate of employee questionnaires was 86.7%. 198 questionnaires were issued to human resource managers, and 141 sets of valid questionnaires were recovered, with an effective recovery rate of 71.7%. At the employee level, there were 712 males, accounting for 54.4%; 598 women, accounting for 45.6%, were equal in gender; Young people and middle–aged and young people, mainly concentrated in 26–30 years old and 31–35 years old, accounting for 33.1% and 24.5% respectively, followed by 21–25 years old, a total of 234 people, accounting for 17.7%; In terms of education, 412 people had vocational or junior college degrees, accounting for 31.5%, and 489 people had full–time bachelor degrees, accounting for 37.3%. 10.9% had high school or technical secondary education, 15.4% had adult undergraduate education, and 4.9% had postgraduate education. In terms of working years, the average tenure was 67.88 months (about 5.5 years), and the longest tenure was 449 months (about 37 years).

## 3.3 Measurement of variables

This paper draws on scales that have been established and validated both nationally and internationally, and the questionnaire is based on scales provided in related articles. To ensure the reliability and validity of the variables in these mature scales in the Chinese context, the original scale was modified and adjusted in strict accordance with the translation–back translation procedure, to ensure the consistency and accuracy of each item in the connotation of the scale. All variables were scored using the Likert scale 7 point scoring method, and scored from "1–7", where "1" represents "completely inconsistent" and "7" represents "completely consistent". The measured value of the variable was measured by the

average score of each item under the variable, this refers to scale scoring. The higher the value, the stronger the performance of the individual or organization on the variable.

Work–family conflict (WFC) was measured using the 4–item scale developed by Joseph et al. (2006) [49]. Cronbach's alpha coefficient was 0.912; Employees' proactive behavior (PB) was measured by the 7–item scale developed by Griffin et al. (2007) [19]. Cronbach's alpha coefficient of 0.932. Emotional exhaustion (EE) was measured by the MBI–GS scale, which is widely used in general industries based on the emotional exhaustion scale developed by Maslach and Jackson (1981)[25]. In combination with domestic scholars Li Chao Ping and Shi Kan (2003) [50], items of the MBI–GS scale were screened according to the more characteristic management situation and cultural characteristics of China, and the 6–item emotional exhaustion scale was finally formed. Cronbach's alpha coefficient was 0.916. The measurement scale of the High Commitment Work System (HCWS) was based on the 10–item scale developed by Xiao and Tsui (2007) in the Chinese context [51], which was split into a 17–item mature scale considering the inconvenience of placing some of these expressions in a single item, with a Cronbach's alpha coefficient of 0.908. According to the demographic characteristics variables proposed by previous studies that may be related to the subject of this study to some extent, this paper mainly selects employee Gender, Age, Education, and Length of Service as the control variables.

According to Kreft (1996), in cross–level studies, it is suggested that at least 30 groups be divided, with each group containing at least 30 samples [52]. In the research of this paper, since the sample size is large enough (1,354 employees) the number of groups is sufficient (141 groups), and the reliability within each group is all greater than 0.7, the sample size of each group can be appropriately reduced. Therefore, in this paper, there are approximately 10 samples in each group.

## 4. Research results

### 4.1 Sample size estimation

According to the research design, this paper is a correlational study that uses structural equation modeling to examine the relationships among variables. Based on this, the sample size estimation was calculated using G*Power 3.1 software. Firstly, in terms of the selection of effect size, Cohen (2013) [53] provided three standards: small ($f^2 = 0.02$), medium ($f^2 = 0.15$), and large ($f^2 = 0.35$). Considering the high correlations between the independent variable work-family conflict, emotional exhaustion, and high commitment work system) and the outcome variable(employees' proactive behavior), indicating a significant influence of the independent variables on the dependent variable, a large effect size ($f^2 = 0.35$) was chosen as one of the input parameters for the calculation.Secondly, following the conventions in psychology and organizational behavior research, a significance level of $\alpha = 0.05$ and a statistical power of $1 - \beta = 0.8$ were selected as the other two estimation parameters. Additionally, when estimating the model complexity, four control variables (i.e., gender, age, education level, and years of work experience) were included in addition to the four independent variables specified in the research hypotheses. Therefore, the number of predictor variables was approximately between 6 and 8. Based on these parameters, I used the "F tests" → "Linear multiple regression: Fixed model, R2 deviation from zero" module in G*Power software for the calculation. The results showed that when the effect size $f^2 = 0.35$, the significance level $\alpha = 0.05$, the statistical power $1 - \beta = 0.8$, and the number of predictor variables was 7, the minimum required sample size was 49. However, the actual sample size was 141, which far exceeded the requirements for medium to large effect sizes, indicating that the sample size was sufficient.

### 4.2 Reliability analysis

To assess the fundamental components of the scale's items and the data distribution, descriptive statistics are used to the scale's items in this study. These statistics mostly contain mean value, standard deviation, skewness, kurtosis, and other information.

Tables 1 and 2 demonstrate how the statistical analysis results of the data of the different questionnaire questions—such as the number of cases, lower, maximum, mean, standard deviation, skewness, and kurtosis—are used to confirm whether

**Table 1. Descriptive statistics of individual variables.**

|  | N | Minimum | Maximum | Mean | Std. Deviation | Skewness | | Kurtosis | |
|---|---|---|---|---|---|---|---|---|---|
|  | Statistic | Statistic | Statistic | Statistic | Statistic | Statistic | Std. Error | Statistic | Std. Error |
| PB1 | 1222 | 1.00 | 7.00 | 5.6465 | 1.21595 | −0.911 | 0.070 | 0.618 | 0.140 |
| PB2 | 1222 | 1.00 | 7.00 | 5.6849 | 1.19228 | −0.957 | 0.070 | 0.751 | 0.140 |
| PB3 | 1221 | 1.00 | 12.00 | 5.5111 | 1.21975 | −0.551 | 0.070 | 0.875 | 0.140 |
| PB4 | 1220 | 1.00 | 45.00 | 5.3861 | 1.66988 | 10.693 | 0.070 | 259.064 | 0.140 |
| PB5 | 1222 | 1.00 | 7.00 | 5.3617 | 1.23494 | −0.674 | 0.070 | 0.306 | 0.140 |
| PB6 | 1222 | 1.00 | 7.00 | 5.3838 | 1.26277 | −0.567 | 0.070 | −0.045 | 0.140 |
| PB7 | 1222 | 1.00 | 7.00 | 5.0565 | 1.32546 | −0.410 | 0.070 | −0.055 | 0.140 |
| WFC1 | 1223 | 1.00 | 7.00 | 3.2739 | 1.67763 | 0.192 | 0.070 | −0.868 | 0.140 |
| WFC2 | 1221 | 1.00 | 7.00 | 2.9984 | 1.72589 | 0.387 | 0.070 | −0.965 | 0.140 |
| WFC3 | 1221 | 1.00 | 7.00 | 3.0901 | 1.69621 | 0.314 | 0.070 | −0.934 | 0.140 |
| WFC4 | 1222 | 1.00 | 7.00 | 3.4984 | 1.76959 | 0.128 | 0.070 | −0.975 | 0.140 |
| EE1 | 1221 | 1.00 | 7.00 | 3.0262 | 1.60919 | 0.327 | 0.070 | −0.819 | 0.140 |
| EE2 | 1218 | 1.00 | 7.00 | 2.8867 | 1.65176 | 0.434 | 0.070 | −0.868 | 0.140 |
| EE3 | 1221 | 1.00 | 7.00 | 3.4791 | 1.68861 | 0.068 | 0.070 | −0.834 | 0.140 |
| EE4 | 1223 | 1.00 | 7.00 | 2.8111 | 1.63267 | 0.485 | 0.070 | −0.785 | 0.140 |
| EE5 | 1221 | 1.00 | 7.00 | 3.1286 | 1.66812 | 0.283 | 0.070 | −0.831 | 0.140 |
| EE6 | 1222 | 1.00 | 35.00 | 2.7987 | 1.86754 | 4.527 | 0.070 | 71.052 | 0.140 |
| Valid N (listwise) | 1197 |  |  |  |  |  |  |  |  |

**Table 2. Descriptive statistics of organizational variables.**

|  | N | Minimum | Maximum | Mean | Std. Deviation | Skewness | | Kurtosis | |
|---|---|---|---|---|---|---|---|---|---|
|  | Statistic | Statistic | Statistic | Statistic | Statistic | Statistic | Std. Error | Statistic | Std. Error |
| HCWS1 | 140 | 1.00 | 7.00 | 4.986 | 1.319 | −0.584 | 0.205 | 0.363 | 0.407 |
| HCWS2 | 141 | 2.00 | 7.00 | 5.667 | 1.138 | −0.907 | 0.204 | 0.972 | 0.406 |
| HCWS3 | 141 | 1.00 | 7.00 | 5.213 | 1.468 | −0.925 | 0.204 | 0.625 | 0.406 |
| HCWS4 | 141 | 1.00 | 7.00 | 5.262 | 1.291 | −0.624 | 0.204 | 0.039 | 0.406 |
| HCWS5 | 141 | 2.00 | 7.00 | 6.028 | 1.189 | −1.297 | 0.204 | 1.111 | 0.406 |
| HCWS6 | 141 | 1.00 | 7.00 | 5.043 | 1.253 | −0.280 | 0.204 | −0.080 | 0.406 |
| HCWS7 | 141 | 1.00 | 47.00 | 4.660 | 3.933 | 8.975 | 0.204 | 97.156 | 0.406 |
| HCWS8 | 141 | 1.00 | 7.00 | 5.404 | 1.320 | −1.122 | 0.204 | 1.649 | 0.406 |
| HCWS9 | 141 | 1.00 | 7.00 | 4.326 | 1.722 | −0.356 | 0.204 | −0.765 | 0.406 |
| HCWS10 | 140 | 1.00 | 7.00 | 4.571 | 1.710 | −0.535 | 0.205 | −0.556 | 0.407 |
| HCWS11 | 141 | 1.00 | 7.00 | 4.993 | 1.301 | −0.638 | 0.204 | 0.253 | 0.406 |
| HCWS12 | 140 | 1.00 | 7.00 | 3.207 | 2.037 | 0.419 | 0.205 | −1.179 | 0.407 |
| HCWS13 | 140 | 1.00 | 7.00 | 4.793 | 1.548 | −0.498 | 0.205 | −0.370 | 0.407 |
| HCWS14 | 141 | 1.00 | 7.00 | 4.801 | 1.648 | −0.502 | 0.204 | −0.726 | 0.406 |
| HCWS15 | 141 | 1.00 | 7.00 | 5.085 | 1.592 | −0.830 | 0.204 | −0.039 | 0.406 |
| HCWS16 | 141 | 1.00 | 7.00 | 4.773 | 1.671 | −0.669 | 0.204 | −0.337 | 0.406 |
| HCWS17 | 141 | 1.00 | 7.00 | 5.638 | 1.348 | −1.088 | 0.204 | 1.089 | 0.406 |
| Valid N (listwise) | 137 |  |  |  |  |  |  |  |  |

**Table 3. Reliability statistics.**

| Variables | Cronbach's Alpha | N of Items |
|---|---|---|
| Z–HCWS | 0.908 | 17 |
| Y–PB | 0.932 | 7 |
| X–WFC | 0.909 | 4 |
| M–EE | 0.929 | 6 |

**Table 4. Comparison and analysis of multi–factor models.**

| Fitting index | CMIN | DF | CMIN/DF | RMSEA | TAG | CFI | SRMR within group | SRMR Inter–group |
|---|---|---|---|---|---|---|---|---|
| Acceptable range | | | 1–5 | < 0.08 | > 0.8 | > 0.8 | < 0.1 | < 0.1 |
| 4–factor model | 1746.839 | 544 | 3.211 | 0.043 | 0.872 | 0.884 | 0.039 | 0.098 |
| 3–factor model | 3543.797 | 549 | 6.455 | 0.068 | 0.684 | 0.711 | 0.071 | 0.108 |
| 2–factor model | 3836.404 | 551 | 6.963 | 0.071 | 0.654 | 0.683 | 0.086 | 0.322 |
| 1–factor model | 7373.795 | 554 | 13.310 | 0.101 | 0.286 | 0.342 | 0.254 | 0.220 |

4–factor model: WFC, EE, HCWS, PB

3 factor model: WFC＋EE, HCWS, PB

2–factor model: WFC＋EE+HCWS, PB

1–factor model: WFC＋EE+HCWS+PB

the survey data follow a normal distribution. The following analysis will be significantly impacted by whether the data is normally distributed. According to Kline (1998), samples are said to follow the normal distribution when the absolute values of skewness and kurtosis are less than three and ten, respectively. The table's formal sample findings demonstrate that each topic's absolute skewness and kurtosis values are fewer than three and ten, respectively. Each topic can follow a normal distribution as skewness and kurtosis both satisfy the requirements of a normal distribution. Reliability and validity analyses, among other statistical analyses, can be performed directly using the data obtained from the questionnaire.

The consistency of the questionnaire research variables on each measurement item was examined in this work using Cronbach's Alpha reliability coefficient to guarantee the efficacy of model fit evaluation and hypothesis testing. The dependability of the scale is deemed good if the Cronbach's Alpha coefficient is higher than 0.7. Table 3 displays the findings of the reliability analysis. The above table shows that the Cronbach's Alpha values for the study's high–level work system, employee initiative behavior, family–work conflict, and emotional exhaustion were 0.908, 0.932, 0.909, and 0.929, respectively. All of these values were higher than 0.7, indicating that the data was reliable.

### 4.3 Validity analysis

By comparing multi–factor models, this study can determine the optimal model for validity analysis. MPLUS 6 is used for analysis, and the research findings are displayed in Table 4 below to confirm that key variables have strong structural validity: CMIN/DF is 3.211 lower than 5, TLI and CFI both reach the standard above 0.8, RMSEA is 0.043 less than 0.08, intra–SRMR is 0.039 less than 0.1, and inter–SRMR is 0.098 less than 0.1—all of which are within the optimum range. The 4–factor model is the best. It demonstrates the superiority of the 4–factor model.

### 4.4 Correlation analysis

This paper statistically examined the association, mean, and standard deviation of work–family ties, employee active behavior, and emotional weariness. As control variables, a few demographic factors were also included in the scale,

**Table 5. Correlation factor analysis of major variables.**

| | Mean value | Standard Deviation | Gender | Age | Education | Length of Service | WFC | EE | HCWS | PB |
|---|---|---|---|---|---|---|---|---|---|---|
| **Gender** | 1.451 | 0.498 | 1 | | | | | | | |
| **Age** | 3.757 | 1.456 | 0.109 *** | 1 | | | | | | |
| **Education** | 2.938 | 1.159 | −0.027 | 0.236 *** | 1 | | | | | |
| **Length of Service** | 3.728 | 1.086 | −0.040 | 0.546 *** | 0.210 *** | 1 | | | | |
| **WFC** | 3.215 | 1.521 | 0.081 ** | 0.017 | 0.041 | 0.112 *** | 1 | | | |
| **EE** | 3.022 | 1.424 | 0.065 * | −0.004 | 0.082 ** | 0.055 | 0.633 *** | 1 | | |
| **HCWS** | 4.933 | 1.056 | 0.060 * | 0.006 | 0.092 ** | 0.011 | 0.028 | −0.007 | 1 | |
| **PB** | 5.433 | 1.064 | −0.020 | 0.044 | −0.054 | −0.010 | 0.181 *** | 0.279 *** | 0.059 * | 1 |

*$p < 0.05$

**$p < 0.01$

***$p < 0.001$

**Table 6. Zero model test of emotional exhaustion and employee active behavior.**

| | Variance between groups t00 | Intra–group variance σ2 | ICC |
|---|---|---|---|
| **EE** | 0.467 | 1.522 | 0.236 |
| **PB** | 0.297 | 0.817 | 0.267 |

including age, gender, degree of education, length of employment with the organization, etc. Table 5 displays the particular outcomes.

The aforementioned table shows that work–family conflict and emotional exhaustion have a positive correlation ($r = 0.633$, $p < 0.01$), while work–family conflict and employees' active behavior have a negative correlation ($r = -0.181$, $p < 0.01$), and emotional exhaustion has a negative correlation ($r = -0.279$, $p < 0.01$), indicating that the aforementioned hypothesis can be confirmed and supported.

## 4.5 Zero model test

The zero model is the premise of multi–layer linear model analysis, which must show that there is variation in M and Y at both the individual level and the team level. Therefore, in this paper, emotional exhaustion and employee active behavior were tested by the zero model. The results are shown in Table 6 below:

Table 4 above illustrates this: The intra–group correlation coefficient (ICC) is 0.236, the intra–group variance2 (σ) is 1.522, and the inter–group variance (t00) is 0.467 for emotional tiredness. Since it makes up 23.6% of the overall variation, the inter–group variation cannot be disregarded. The intra–group variance 2 σ is 0.817 and the inter–group variance t00 is 0.297 for employees' active behavior. Indicating that emotional exhaustion and employees' active behavior also had team–level variables that could explain the individual–level variables, the intra–group correlation coefficient (ICC) was 0.267, the inter–group variation could not be disregarded, and the inter–group variation accounted for 26.7% of the total variation.

## 4.6 Hypotheses testing

In this paper, Gender, Age, Education, and Length of Service were used as control variables at the employee level, family–work conflict (WFC) was used as an independent variable at the individual level, and high commitment work system (HCWS) was used as a regulatory variable at the organizational level. Emotional exhaustion (EE) is the intermediary

**Table 7. Path coefficients of MPLUS cross–layer analysis.**

|  |  | EE | PB |
|---|---|---|---|
|  |  | Model 1 | Model 2 |
| In–group variables | Gender | −0.036 | −0.096 |
|  | Age | 0.002 | 0.002 |
|  | Education | 0.011 ** | 0.001 |
|  | Length of Service | 0.002 *** | −0.001 |
|  | WFC | 0.540 *** | −0.020 |
|  | EE |  | 0.138 *** |
| Inter-group variables | Z |  | 0.065 |
|  | MZ |  | 0.065 * |
| Stats | Variance between groups t00 | 0.217 | 0.241 |
|  | Intra–group variance σ2 | 0.967 | 0.716 |

**Table 8. Analysis of mediating effect.**

|  | Coefficient | T | P | Lower interval | Upper interval |
|---|---|---|---|---|---|
| Direct effects | −0.020 | −0.581 | 0.561 | −0.086 | 0.047 |
| Indirect effects | −0.075 | −4.361 | 0.000 | −0.108 | −0.041 |
| Total effect | −0.094 | −3.074 | 0.002 | −0.154 | −0.034 |

variable at the individual level, and employee active behavior (PB) is the dependent variable at the individual level for SEM path analysis, as shown in the following table (Table 7).

From the above table, model 1 of this paper shows that family–work conflict has a significant positive effect on emotional exhaustion (β=0.540, p<0.001), while family–work conflict has no significant direct negative effect on active work behavior (β=-0.020, p>0.05).Emotional exhaustion has a significant negative impact on employees' proactive behavior (β=-0.138, p<0.001). Hypothesis 1 and hypothesis 2 are valid.

In addition, this paper uses the Bootstrap sampling test method to study the mediating effect, and the specific effect value results are shown in Table 8.

The mediating effects test in the above table shows that: Family–work conflict has a direct effect of -0.020, 95% CI [-0.086, 0.047], which includes 0, indicating that the direct effect is not significant. The indirect effect of family–work conflict on employees' proactive behavior through emotional exhaustion is -0.075, 95% CI [-0.108, -0.041], which does not contain 0, indicating that the indirect effect is significant. This suggests that emotional exhaustion acts as a mediator between work–family conflict and employees' proactive behavior. The initial effect of work–family conflict is emotional weariness, which in turn influences employees' active conduct, hypothesis 2 is valid.

To test hypotheses 3 and 4. In this paper, MPLUS software was used to analyze the data. The path coefficient table showed that the coefficient of emotional exhaustion and high commitment work system was -0.065, which reached a significant level at 0.05, indicating a positive moderating effect (Table 9). The specific effect was shown in the following table, and the level of different high commitment work systems was different (Fig 2). When the work system level is high in commitment, the emotional exhaustion variable has a significant negative effect on the employee's active behavior (β=-0.207, P<0.001), and when the work system level is low in commitment, the emotional exhaustion variable has no significant negative effect on the employee's active behavior (β=-0.069, p>0.05), and the difference between the two is -0.138. P<0.05, indicating a moderating effect, hypothesis 3 is valid.

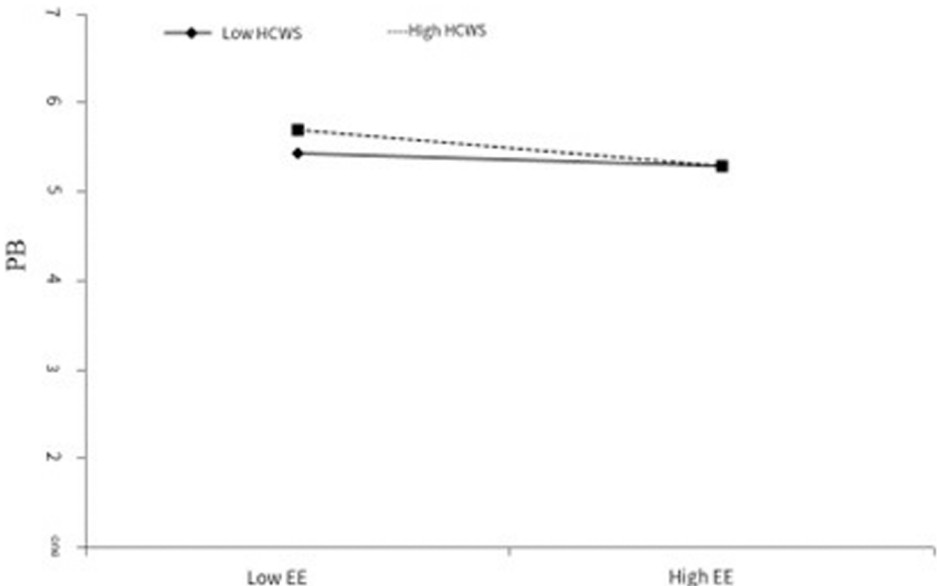

**Fig 2. The modulation of high commitment work systems between emotional exhaustion and active behavior.**

**Table 9. Simple slope of adjustment analysis.**

| Adjusting Variables | Beta. | SE | T | P |
|---|---|---|---|---|
| Low high commitment work system | -0.069 | 0.044 | -1.562 | 0.118 |
| High commitment work system | -0.207 | 0.044 | -4.715 | 0.000 |
| Difference | -0.138 | 0.060 | -2.316 | 0.021 |

**Table 10. Moderated mediators.**

| Moderating variables | Standardization coefficient | SE | T | P | Lower interval | Upper interval |
|---|---|---|---|---|---|---|
| Low packet | −0.037 | 0.024 | −1.579 | 0.114 | −0.083 | 0.009 |
| High Grouping | −0.112 | 0.024 | −4.728 | 0.000 | −0.158 | −0.065 |
| Difference | −0.075 | 0.033 | −2.290 | 0.022 | −0.139 | −0.011 |

To further study whether the high commitment work system can regulate the mediating influence of emotional exhaustion on the mechanism of family–work conflict on employees' active behavior, a moderated mediating effect test was conducted on the data. The test results are shown in Table 10:

As can be seen from Table 10, the mediating effect of emotional exhaustion in the high commitment work system has a difference of -0.075, 95% confidence interval [-0.139,-0.011], excluding 0, indicating that the mediating effect is moderated, and when the adjustment becomes a high grouping, the mediating effect is -0.112, 95% confidence interval is [-0.158, -0.065], does not contain 0, indicating that the mediating effect is significant; when the moderating variable is low, the mediating effect is -0.037, 95% confidence interval is [-0.083, 0.009], including 0, indicating that the mediating effect is not significant; the mediating effect of high Z is higher than that of low Z. This suggests a moderation effect of the high commitment work system on the mediation process. In economic terms, as organizations adopt a high commitment work

**Table 11. Influence analysis of employees' proactive behavior by gender.**

| | Gender (mean ± standard deviation) | | t | p |
|---|---|---|---|---|
| **Employees' Proactive Behavior** | 1.0(n = 712) | 0.0(n = 598) | | |
| | 5.440 ± 1.090 | 5.400 ± 1.050 | 2.580 | 0.312 |

**Table 12. Analysis of the Influence of Employees' Age on Proactive Behavior.**

| Age (mean ± standard deviation) | | | | | | | | | t | p |
|---|---|---|---|---|---|---|---|---|---|---|
| **Employees' Pro-active Behavior** | 1.0(n = 3) | 2.0(n = 234) | 3.0(n = 437) | 4.0(n = 324) | 6.0(n = 110) | 7.0(n = 44) | 8.0(n = 18) | 9.0(n = 7) | 10.0(n = 2) | |
| | 5.44 ± 1.09 | 5.38 ± 1.04 | 5.41 ± 1.02 | 5.34 ± 1.10 | 5.51 + 1.110 | 5.46 ± 1.03 | 5.71 ± 1.07 | 4.71 ± 1.36 | 4.71 ± 1.72 | 1.027 | 0.415 |

system, the mediation of work–family conflict on proactive behavior through employee emotional exhaustion diminishes, whereas its influence intensifies under the opposite scenario. Hypothesis 4 is valid.

### 4.7 Analysis of the impact of control variables

**4.7.1 Influence of gender on employees' proactive behavior.** An independent samples t–test (Ruxton, 2010) was used to determine whether there is a significant difference in the proactive behavior of employees based on gender. In this paper, employees were split into two groups based on gender, with the male group being assigned a value of 1 and the female group being assigned a value of 0. Influence variations. The table below illustrates that there is no difference between the various gender samples for employees' proactive behavior, as none of the samples will exhibit significance for this behavior ($p > 0.05$) (Table 11).

**4.7.2 Influence of age on employees' proactive behavior.** Employees were divided into the following age groups in this paper: (1) 20 years or less; (2) 21- 25 years; (3) 21- 30 years; (4) 31- 35 years; (5) 36- 40 years; (6) 41- 45 years; (7) 46- 50 years; (8) 51- 55 years; (9) 56- 60 years; and (10) 60 years or more. The one–way ANOVA method (Bewick, Cheek, and Ball, 2004) was used to determine whether there was a significant difference in the impact of age on employees' proactive behavior (Table 12).

From the above table, it can be seen that: none of the different age samples will show significance for Employees' proactive Behavior ($p > 0.05$), which means that there is no difference between the different age samples for Employees' proactive Behavior.

**4.7.3 The effect of length of service in the enterprise on employees' proactive behavior.** In this paper, employees were categorized according to their age as (1) 60 months and below, (2) 60–120 months, (3) 120–180, and (4) 180 months and above, and were divided into groups according to '1-4', therefore, one–way ANOVA method (Bewick, Cheek and Ball, 2004) was applied to analyze whether there is a significant difference in the effect of age on proactive behavior in our company. To analyze whether there is a significant difference in the effect of length of service on Employees' proactive Behavior in this enterprise. As can be seen from the table below: none of the samples with different lengths of service in this company will show significance ($p > 0.05$) on Employees' proactive Behavior, which means that there is no difference in Employees' proactive Behavior between the samples with different length of service in this company (Table 13).

**4.7.4 The effect of education level on employees' proactive behavior.** In this paper, employees were classified according to their age into (1) high school–secondary school; (2) vocational high school–college; (3) adult bachelor's degree; (4) full–time bachelor's degree; (5) master's degree; and (6) doctoral degree and divided into groups according to '1-6', therefore, the one–way ANOVA method (Bewick, Cheek and Ball, 2004) to analyze whether there is a significant difference in the influence of educational attainment on Employees' proactive Behavior.

**Table 13. Analysis of the impact of employees' proactive behavior due to their length of service in their own company.**

| | Length of service in the enterprise (mean±standard deviation) | | | | t | p |
|---|---|---|---|---|---|---|
| employees' proactive Behavior | 1.0(n=704) | 2.0(n=355) | 3.0(n=143) | 4.0(n=35) | | |
| | 5.460±1.020 | 5.380±1.040 | 5.420±1.030 | 4.360±1.320 | 1.264 | 0.465 |

**Table 14. Analysis of the effect of employees' educational attainment on proactive behavior.**

| | Educational level (mean±standard deviation) | | | | | | t-value | p-value |
|---|---|---|---|---|---|---|---|---|
| Employees' proactive Behavior | 1.0(n=143) | 2.0(n=412) | 3.0(n=201) | 4.0(n=489) | 5.0(n=57) | 6.0(n=7) | | |
| | 5.75±0.98 | 5.36±1.06 | 5.45±1.08 | 5.36±1.07 | 5.15±0.98 | 6.23±2.95 | 4.774 | 0.000** |

As can be shown from the above table, there is a significant difference (p<0.05) in the proactive behavior of employees across samples of varying educational levels, indicating that those samples differ from one another (Table 14). There is a more noticeable difference between the group mean score comparison results for "1.0>2.0; 1.0>3.0; 1.0>4.0; 1.0>5.0; 6.0>2.0; 6.0>4.0; 6.0>5.0," i.e., the sample of different education levels has differences for Employees' proactive Behavior. The educational level for Employees' proactive Behavior shows 0.01 level of significance (F=4.774, p=0.000). 5.0', meaning that there are notable variations in the proactive behavior of all employees across various education samples. Furthermore, as education levels rise, the disparity in employees' proactive behavior becomes more pronounced.

## 5 Research conclusions and implications

### 5.1 Main conclusions and contributions

Role conflicts will lead to work–family conflicts since employees play distinct roles in the two work and family settings. These conflicts will also have an impact on the organization's employees' behavior and psychology [54]. The impact of work–family conflict on proactive behavior by employees and its internal influencing mechanism are the main topics of this paper. This paper, which is based on resource conservation theory, investigates the boundary conditions of a high commitment work system and how work–family conflict influences employees' proactive behavior through emotional weariness. The findings indicate that employees who experience work–family conflict will be less proactive. This is because employees will experience unpleasant feelings at work [55], work pressure [56], and a constant depletion of internal resources including personal energy and emotion as a result of the contradiction and conflict between work and family. The notion of resource conservation states that when people's internal resources are inadequate, they are less likely to act kindly and philanthropically [57]. to compensate for or prevent the use of internal resources. Conflict between work and family will therefore lessen employees' proactive conduct. The relationship between work–family conflict and proactive conduct is mediated by emotional exhaustion; work–family conflict will exacerbate emotional weariness in employees. Employees are less likely to take initiative when they are emotionally exhausted. This is due to the fact that work–family conflicts cause employees' emotional resources to be depleted, which leads to a sense of physical and emotional exhaustion at work and certain undesirable moods like burnout and impatience. Employees who experience this type of emotional weariness on a regular basis will therefore be more aware of the benefits of their resource investment, including their own time, effort, physical energy, and other resources invested in their jobs. As a result, this typically lowers employees' proactive activity, which further validates the findings of earlier research from many angles. The hypothesis test results demonstrate that the high commitment work system moderates the relationship between employees' proactive activity and emotional weariness. This is due to the fact that through the implementation of a high commitment work system in human resource management, the company is able to comprehend the actual needs of its workforce and communicate the enterprise vision to them, which they acknowledge, in order to achieve the shared objectives and values of the company and its

employees through emotional commitment [58]. Therefore, when employees feel emotionally exhausted, the high commitment work system allows them to replenish their resources with new ones they value in the company, which lessens their emotional tiredness. At the same time, workers in companies with high commitment work systems will feel more obligated to come back [59], which will lead them to act on their own initiative to learn, grow, and strive for greatness. Therefore, emotional weariness will have less of an effect on employees' proactive conduct in companies that use high commitment work systems.

The following three elements primarily represent this paper's contribution: In addition to offering a theoretical window with some theoretical significance and a novel theory for organizations to encourage employee proactive behavior, this paper first enriches the research perspective on employee proactive behavior by examining the anthems of employee proactive behavior. This paper examines the relationship between work–family conflict and employee proactive behavior from the standpoint of resource conservation theory, which is different from earlier research on the topic from the perspectives of social exchange and social cognition theory [60]. It is simple to overlook the impact of the resource conservation mechanism on employees' proactive attitude and behavior when contrasted with the social exchange mechanism. Thus, the study of this work significantly enhances and broadens the comprehension of resource conservation theory in addition to extending its application scope.

Second, prior research on work–family relationships has focused more on individual attitude and work performance as outcome variables. Studies on work behavior, on the other hand, have primarily focused on counterproductive behavior and organizational citizenship behavior, rarely examining the outcome variable of active behavior. This paper expands the new content and direction of work–family relationship research by presenting the theoretical verification to examine employees' positive behavior performance in the workplace from the perspective of work–family relationships, as seen through the lens of work–family conflict.

Third, the impact of the high commitment work system as the antecedent variable is given more attention in the studies that are currently available on the subject [61]. Additionally, the impact of implementing this mode of human resource management on the organization and its employees is discussed from the viewpoint of the enterprise or human resource managers. The current research findings are still unsatisfactory, despite the fact that some researchers have taken into account the influence on the business and people by using moderating variables such high–performance work systems and human resource management practices. This paper breaks through the existing research on the high commitment work system as the dextral variable by demonstrating that it can govern employee emotional weariness and employee initiative behavior.

## 5.2 Management implications and prospects

According to the research of this article, we can get the following two enlightenments:

(1) Create a positive work–family dynamic. While asking and motivating workers to increase their contributions to the company, companies should also take personal responsibility for their factors, including employees' families. They must to take the effort to assist staff members in managing the work–family relationship, bringing the two together rather than in conflict, enhancing employee satisfaction, and then enhancing initiative in conduct. Businesses should adjust to the times, abandon the outdated management practices and policies that emphasized attempting to control employees' personal lives and hard work, and adopt more flexible work arrangements. Organizations should provide a specified time range rather than a set of tasks to complete at a certain node so that workers can freely and sensibly plan and manage their work and family obligations. Businesses should incorporate all employee traits into organizational and post–design strategies. They can also implement flexible work arrangements, such as job sharing, flexible vacation policies, work–family balance optimization, and remote work. Companies can improve their relationship with employees' families by involving them in their daily work. This will help the family members understand the importance of the employee to the

company, understand pertinent enterprise management policies, remove the detrimental effects of information asymmetry on families, and prevent potential conflicts between families and employees from starting in the first place.

(2) Create a high commitment work environment that keeps up with The Times' evolution. In order to increase employee happiness and satisfaction, the high commitment work system promotes increased trust and resource support, a safe and laid–back organizational environment for employee development through hiring, intensive training, competitive pay, team–oriented performance evaluation, participatory management, and other HRM practices. Encourage staff members to actively absorb the organization's objectives and incorporate them into their own professional aspirations. A new generation of enthusiastic, free–spirited workers now dominates organizations. Organizations must make significant investments in every facet of a high commitment work environment in order to encourage their initiative. Businesses should, first and foremost, adapt to the times, continuously optimize current HRM policies to adapt to the evolving workforce structure, pay attention to the recruitment model based on development potential, pay attention to the training of knowledge and skills in conjunction with competency requirements, implement flexible organizational design, job reference and job design, and encourage employees to actively participate. To encourage employee initiative, strengthen their dedication to the company, and raise awareness of sustainable development and change, pay close attention to the performance review with the employees' sustainable development at its center and the employment policy with a sense of security. Second, depending on the various internal employee states, company managers should implement management rules with varying directions and pertinence. Employers can, for instance, encourage employees with high psychological capital to take more initiative by improving their recognition of the value of their work through job expansion and enrichment, job rotation, job autonomy and flexibility, and other HRM practices. In order to help employees manage their emotions and boost their confidence in their own abilities and organizational status, the company should be adept at recognizing when employees are in a bad mood, helping them to improve it through the employee care mechanism, and providing them with more opportunities to participate in management and decision–making, as well as more extensive training and job rotation.

There are still certain research gaps despite the tremendous efforts made in this paper to examine the relationship between work–family conflict and proactive conduct on the part of employees. This is also the focus of the author's future research.

(1) The relationship between work and family is reciprocal, encompassing both the influence of work on family and the influence of family on work. What effect does family–work conflict have on employees' proactive behavior? This paper primarily examines work–family conflict. Future research should examine the distinctions between these two states and how the equilibrium between them affects proactive behavior. Furthermore, work–family interactions can both strengthen and deteriorate in addition to causing conflict (Zheng Shilin and Xia Fubin, 2017) [62].

(2) Since the majority of the variables suggested in this paper were derived from foreign literature, all of the scales used were created in other nations. Despite their strong validity and reliability, it is still up for debate if they can be fully applied to the Chinese environment. Many scales used to measure constructs need to adapt to the times, especially in light of the shifting labor force composition, the influence of a fresh wave of technological revolution, and the changing times themselves. In order to properly reflect the practices of businesses in the Chinese setting, future paper can locally adjust and optimize the assessment of pertinent variables depending on a particular context. Furthermore, this paper may be impacted by common technique bias because it employed a one–time approach for data collecting and analysis. Additionally, this paper uses cross–sectional data, which may somewhat differ from reality when examining work–family conflict and work–family promotion balance. To guarantee the caliber and dependability of data gathering, longitudinal design—that is, dynamic design in time—can be used in the next phase. The discontinuous stability of work–family conflict and work–family promotion under varying equilibrium degrees may be more consistent in this approach.

## Author contributions

**Conceptualization:** Yule Wan.

**Writing – original draft:** Hongyuan Zhang, Yule Wan.

**Writing – review & editing:** Hongyuan Zhang, Yule Wan, Shuming Zhao.

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
