## [Decision Letter · Decision Letter 0]

18 Nov 2024

PONE-D-24-42561Research on the Influence Mechanism of Work-Family Conflict on Employees' Proactive Behaviors-A Moderated Mediator ModelPLOS ONE

Dear Dr. Wan,

Thank you for submitting your manuscript to PLOS ONE. After careful consideration, we feel that it has merit but does not fully meet PLOS ONE’s publication criteria as it currently stands. Therefore, we invite you to submit a revised version of the manuscript that addresses the points raised during the review process.

1. The paper requires language verification.

2. The representation of key variables should be standardized. For instance, terms such as "employees' proactive behavior," "employee initiative behavior," and "Employee active behavior" should be unified.

3. The dataset for Mplus and factor models is not provided, rendering data availability incomplete.

4. The clarity of the figures is poor; it is recommended that they be replaced according to standard specifications.

5. Regarding the sample size for cross-level analysis, scholars have established clear and stringent criteria. The authors have not disclosed pertinent critical information, preventing us from determining whether these standards are met. Please note that this is extremely important as it impacts the rigor of the research methodology and the scientific validity of the results.

6. It is advised that the authors exercise greater rigor in their manuscript, conducting multiple reviews prior to submission to avoid obvious errors.

We look forward to receiving your revised manuscript.

Kind regards,

Jianpeng Fan

Academic Editor

PLOS ONE

Journal Requirements:

2. Thank you for stating the following financial disclosure: [National Natural Science Foundation of China Special Project: "Research on Digital Human Resource Development and Management within Human-Machine Interaction Scenarios" (7234-2027)�The 6th Phase of the "521 Project" for Science and Technology Funding in Lianyungang City: "Research on the Pathways for Lianyungang to Expeditiously Build a Prominent Modern Ocean Industrial System" (LYG0 6521 202391)]. Please state what role the funders took in the study. If the funders had no role, please state: "The funders had no role in study design, data collection and analysis, decision to publish, or preparation of the manuscript." If this statement is not correct you must amend it as needed. Please include this amended Role of Funder statement in your cover letter; we will change the online submission form on your behalf.

Reviewers' comments:

Reviewer's Responses to Questions

**Comments to the Author**

1. Is the manuscript technically sound, and do the data support the conclusions?

Reviewer #1: Yes

Reviewer #2: No

Reviewer #3: Partly

Reviewer #4: Yes

2. Has the statistical analysis been performed appropriately and rigorously? 

Reviewer #1: Yes

Reviewer #2: No

Reviewer #3: I Don't Know

Reviewer #4: Yes

3. Have the authors made all data underlying the findings in their manuscript fully available?

Reviewer #1: Yes

Reviewer #2: No

Reviewer #3: No

Reviewer #4: Yes

4. Is the manuscript presented in an intelligible fashion and written in standard English?

Reviewer #1: No

Reviewer #2: No

Reviewer #3: No

Reviewer #4: Yes

5. Review Comments to the Author

Reviewer #1: Dear Editor,

Thank you for providing me with the opportunity to review this manuscript. I have read the complete manuscript and found it to be of good quality. However, there is always room for improvement. Below are some changes that need to be incorporated before publication:

The abstract is very brief and not well-written. It should be expanded and improved for clarity.

The introduction is overall well-written and logically coherent. However, it lacks the latest references, which could help strengthen the research gap.

In the literature review, the authors developed strong arguments on," "Research on the Influence Mechanism of Work-Family Conflict on Employees' Proactive Behaviors-A Moderated Mediator Model" However, there are no recent references that support the literature with relevant theories.

The methods section should be clearer to the reader. On what basis were these organizations from Nanjing, Hefei, Nantong, Lianyungang and other regions? This section must be more understandable.

The results section is well-written and well-explained by the authors. The proposed questions are clearly addressed and interpreted effectively.

Limitations and future directions are sound clear.

Reviewer #2: 1. Please provide rationale for the sample using g power as effect size is low.

2.reliabilty statistic must show other measures of normality data

3.provide rationale foe r direct effects not being normal

4.data must be provided fully in repository , for mplus software.

Reviewer #3: - Title: Remove the phrase "(research on)" from the title and adjust writing style for accuracy and conciseness.

Grammar and Language

- Numerous grammatical and word usage errors are present throughout the manuscript. The whole manuscript would benefit from rewording by a native English editor for clarity and improved readability.

Introduction

- First Sentence needs a supporting reference to provide context, similarly to many sentences in the whole manuscript that claim fact without references.

- The phrase "As the main force of high-quality development" is unclear; specify what is being referred to here.

- Paragraph 2: ("When discussing the relevant...") is poorly structured, with overly long sentences. Revise to improve readability and flow.

Theoretical Framework

- Resource Conservation and Social Exchange Theories: The connection between these theories and the study is inadequately established. Strengthen the theoretical alignment to clarify relevance.

- Propositions: Some propositions are ambiguous and may mislead readers, such as the use of "both" in Paragraph 2, which lacks clarity on what it refers to.

Background, Aim, and Motivation

- Provide a clearer explanation (background) of the existing literature and identify the research gap that this study addresses.

- Clearly state the aim of the study within the introduction.

- Explain the key research question and motivation for conducting this study.

Terminology and Consistency

- Avoid inconsistencies by using terms such as "employee initiative behavior" and “individual's initiative” and just use the same term of "proactive behavior” to match the terminology used elsewhere.

Theory and Hypotheses Development

- Subsection 2.2: The statement "Therefore, work-family conflict has a positive impact on emotional exhaustion" is presented as a result without sufficient justification. Clarify why this impact is considered positive and provide a theoretical basis.

- Gendered Language: The manuscript inconsistently uses “he” to refer to individuals, excluding "she." Revise to include gender-neutral language.

- Psychological Resource: The statement "As a psychological resource, positive emotions..." needs clarification. Define what is meant by "psychological resource."

Hypothesis Development

- Hypothesis 3 and 4 (Subsection 2.3) lack a solid theoretical foundation. Expand and substantiate hypotheses with additional references to strengthen theoretical support

Section 3: Research Design

- Participants and Exclusion Criteria:

- Clarification on "Minors": Define what is meant by "minors" and explain why they were excluded from the survey.

- Data Analysis Justification:

- Provide a rationale for selecting Mplus as the data analysis software and explain its suitability for testing the proposed hypotheses.

- Data Collection Method:

- Clarify Sampling Process: Detail how participants were recruited for the study (e.g., through an online survey or other method) to improve transparency.

- Section 3.3: Questionnaire and Measurement Scales:

- Clarify "Mature Internal and External Articles": Explain what is meant by "mature internal and external articles." If this refers to validated sources or established scales, specify this for clarity.

- Measurement Value Explanation: The phrase "The measured value of the variable was the average value of each item" needs further explanation. Specify whether this refers to scale scoring or another calculation.

- Clearly explain what "After Z-HCWS is removed" refers to in the table (1) to avoid confusion.

- The current interpretation lacks discussion of the findings with relating to exist research. Explain the significance of these results in the study’s broader context.

- Clarify Moderation: The phrase "positively moderated" may be misleading without an explanation. Define how moderation affects the results and consider including effect sizes to convey the magnitude of differences.

- The in-text citations have not been correctly inserted. Please follow the appropriate reference style.

..............

The paper over is promising, and the topic is interesting. However, the article needs further development.

Reviewer #4: First of all, I would like to thank the authors for their efforts as well as for the opportunity to review the manuscript, "Research on the Influence Mechanism of Work-Family Conflict on Employees' Proactive Behaviors—A Moderated Mediator Model."

- The abstract is well-written and effectively represents the work that has been conducted.

- The research context is clearly identifiable.

- The theoretical elements are well introduced with appropriate bibliographic sourcing.

- The methodology used is relevant.

- The measurement tools employed are suitable, and the psychometric metrics are satisfactory.

- The discussion elements are well-structured and written, allowing for practical insights to be drawn from the results obtained.

1/ In the introduction, there are some instances of typos, for example, “job characteristics[3]between employees and organizations[4]few scholars have also[5].”

2/ Generally, it seems that different citation methods have been used throughout the manuscript.

3/ The manuscript would benefit from simplifying certain introductory elements, particularly section 2.2, “The mediating effect of emotional exhaustion on employees' proactive behavior,” whose extensiveness may lead to confusion. Therefore, I kindly ask you to consider reducing the density of this section for clarity.

4/ In the methods section, the manuscript would benefit from standardizing the way numerical data is reported (e.g., “26.92 percent” and “34.62%” later on).

5/ In the results section, I suggest considering standardizing the data reporting and providing an intermediate conclusion at the end of each individual analysis, so that readers can follow the authors' reasoning without needing to go back to the introduction section to read the hypotheses.

6. PLOS authors have the option to publish the peer review history of their article (what does this mean? ). If published, this will include your full peer review and any attached files.

**Do you want your identity to be public for this peer review?** For information about this choice, including consent withdrawal, please see our Privacy Policy .

Reviewer #1: No

Reviewer #2: **Yes: ** yumna ali

Reviewer #3: **Yes: ** Badr Albaram

Reviewer #4: **Yes: ** Alizée Poli

---

## [Author Response · Author response to Decision Letter 1]

16 Dec 2024

Dear PLoS ONE Publisher, Editor, and Reviewers

I would like to express my sincere gratitude to the reviewers and experts for their valuable comments on the article "The Influence Mechanism of Work-Family Conflict on Employees' Proactive Behaviors-Moderated Mediator Model". Based on the comments and suggestions of the reviewers, the author has made revisions to the article, and I will explain each revision in turn below:

1. The paper requires language verification.

Proofread and modify the language expression of the whole paper according to the modification opinions, ensuring that the grammar is correct and the language logic is clear. Since not being a native English speaker may cause some problems in language expression, the author and the team decided to review the feedback of the second time and ask professionals to modify it after modification. I hope experts and editors will forgive me.

2. The representation of key variables should be standardized. For instance, terms such as "employees' proactive behavior," "employee initiative behavior," and "Employee active behavior" should be unified.

According to the modification opinions, each variable in the whole paper is expressed in uniform professional terms. Unified use:Work-family Conflict; Employees' proactive Behavior; High commitment Work System; Emotional Exhaustion

3. The dataset for Mplus and factor models is not provided, rendering data availability incomplete.

The data sets for Mplus and factor models are uploaded in the supporting files.

4. The clarity of the figures is poor; it is recommended that they be replaced according to standard specifications.

The images have been adjusted to the definition required by the journal and re-uploaded.

5. Regarding the sample size for cross-level analysis, scholars have established clear and stringent criteria. The authors have not disclosed pertinent critical information, preventing us from determining whether these standards are met. Please note that this is extremely important as it impacts the rigor of the research methodology and the scientific validity of the results.

An expression about sample size selection has been added in section 3.3 of the article According to Kreft (1996), in cross-level studies, it is suggested that at least 30 groups be divided, with each group containing at least 30 samples. In the research of this paper, since the sample size is large enough (1,354 employees) and the number of groups is sufficient (141 groups), and the reliability within each group is all greater than 0.7, the sample size of each group can be appropriately reduced. Therefore, in this paper, there are approximately 10 samples in each group.

6. It is advised that the authors exercise greater rigor in their manuscript, conducting multiple reviews prior to submission to avoid obvious errors.

According to the opinions of experts and editors, the paper has been comprehensively modified. The revised part is marked and uploaded. See revised version for details.

Reviewer #1: Dear Editor,

Thank you for providing me with the opportunity to review this manuscript. I have read the complete manuscript and found it to be of good quality. However, there is always room for improvement. Below are some changes that need to be incorporated before publication:

1.The abstract is very brief and not well-written. It should be expanded and improved for clarity.

According to the revised opinions, the content of the abstract is expanded and the content of the research significance is added. See the last two sentences of the abstract in the article:This article contributes to the study of work-family conflict and employee proactive behavior, enriching the existing research and helping to better understand the relationship between work-family conflict and employee proactive behavior. It also provides suggestions for effective work-family conflict management strategies for enterprises, which can help to enhance employee work performance and achieve enterprise goals.

2.The introduction is overall well-written and logically coherent. However, it lacks the latest references, which could help strengthen the research gap.

According to expert opinions, references in recent years are added in the introduction as support to enrich the content of the introduction. See the introduction on page 2.

3.In the literature review, the authors developed strong arguments on," "Research on the Influence Mechanism of Work-Family Conflict on Employees' Proactive Behaviors-A Moderated Mediator Model" However, there are no recent references that support the literature with relevant theories.

In Section 2.1, some relevant literatures in recent years are added as support.

4.The methods section should be clearer to the reader. On what basis were these organizations from Nanjing, Hefei, Nantong, Lianyungang and other regions? This section must be more understandable.

The paper uses data from multiple places and various fields, and the sample size is large enough. In order to avoid ambiguity among readers, the author decides to delete relevant expressions.

The results section is well-written and well-explained by the authors. The proposed questions are clearly addressed and interpreted effectively.

Limitations and future directions are sound clear.

Reviewer #2: 1. Please provide rationale for the sample using g power as effect size is low.

The selection principle of sample size is supplemented according to the revision suggestions. For details, see the last paragraph of section 3.3 of the article.

2.reliabilty statistic must show other measures of normality data.

According to the revised suggestions, the table is added in the article: Descriptive Statistics of individual variables and Descriptive Statistics of organizational variables are detailed in Section 4.1.

3.provide rationale for direct effects not being normal

The article adds the relevant criteria for data determination, see section 4.3.

4.data must be provided fully in repository , for mplus software.

Data sets on Mplus and factor models are uploaded in the support file.

Reviewer #3: - Title: Remove the phrase "(research on)" from the title and adjust writing style for accuracy and conciseness.

The "research on" in the title has been deleted according to the modification suggestions. Change The title to "The Influence Mechanism of Work-Family Conflict on Employees' Proactive Behaviors-A Moderated Mediator Model".

Grammar and Language

- Numerous grammatical and word usage errors are present throughout the manuscript. The whole manuscript would benefit from rewording by a native English editor for clarity and improved readability.

Proofread and modify the language expression of the whole paper according to the modification opinions, ensuring that the grammar is correct and the language logic is clear. Since not being a native English speaker may cause some problems in language expression, the author and the team decided to review the feedback of the second time and ask professionals to modify it after modification. I hope experts and editors will forgive me.

Introduction

- First Sentence needs a supporting reference to provide context, similarly to many sentences in the whole manuscript that claim fact without references.

References have been added to the first sentence of the introduction according to the revised suggestions, and references have also been added to other parts of the article.

- The phrase "As the main force of high-quality development" is unclear; specify what is being referred to here.

To change this sentence to "Enterprises have played an important role in promoting high-quality development", see the introduction section, page 2 of the paper.

- Paragraph 2: ("When discussing the relevant...") is poorly structured, with overly long sentences. Revise to improve readability and flow.

According to the suggestions, the grammar and expression of the whole text are revised.

Theoretical Framework

- Resource Conservation and Social Exchange Theories: The connection between these theories and the study is inadequately established. Strengthen the theoretical alignment to clarify relevance.

By adding references, restating relevant content, establishing the connection between resource conservation theory and research, and deleting the social exchange theory, the research team found through discussion that the social exchange theory was not enough to support the existing inferences, so the social exchange theory was deleted.

- Propositions: Some propositions are ambiguous and may mislead readers, such as the use of "both" in Paragraph 2, which lacks clarity on what it refers to.

Relevant propositions are reexpressed, as shown in the highlight of article 2 Theoretical basis and research hypothesis.

Background, Aim, and Motivation

- Provide a clearer explanation (background) of the existing literature and identify the research gap that this study addresses.

- Clearly state the aim of the study within the introduction.

- Explain the key research question and motivation for conducting this study.

Rewrite the last paragraph of the introduction to clarify the purpose, significance and problem to be solved. For details, see the last paragraph of Article 1 Introduction.

Terminology and Consistency

- Avoid inconsistencies by using terms such as "employee initiative behavior" and “individual's initiative” and just use the same term of "proactive behavior” to match the terminology used elsewhere.

Uniform terminology used throughout the text. Unified use: Work-family Conflict; Employees' proactive Behavior; High commitment Work System; Emotional Exhaustion

Theory and Hypotheses Development

- Subsection 2.2: The statement "Therefore, work-family conflict has a positive impact on emotional exhaustion" is presented as a result without sufficient justification. Clarify why this impact is considered positive and provide a theoretical basis.

In 2.2 The mediating effect of emotional exhaustion on employees' proactive behavior, theoretical derivation and literature citations are added to propose the positive effects of work-family conflict on emotional exhaustion.

- Gendered Language: The manuscript inconsistently uses “he” to refer to individuals, excluding "she." Revise to include gender-neutral language.

According to expert opinion, the full text uses the uniform "he"

- Psychological Resource: The statement "As a psychological resource, positive emotions..." needs clarification. Define what is meant by "psychological resource."

Considering that psychological resources are not the content of this paper, the language expression of 2.2 is reorganized, and the expression about psychological capital is deleted.

Hypothesis Development

- Hypothesis 3 and 4 (Subsection 2.3) lack a solid theoretical foundation. Expand and substantiate hypotheses with additional references to strengthen theoretical support

Section 2.3 is reformulated and the reference material is added to strengthen the theoretical support, see article Section 2.3 for details

Section 3: Research Design

- Participants and Exclusion Criteria:

- Clarification on "Minors": Define what is meant by "minors" and explain why they were excluded from the survey.

In Article 3.1, I define "Minors" and explain why they are excluded from the survey. “Since this survey is for employees of enterprises, it is illegal for enterprises to hire employees under the age of 18 in China, so minors (less than 18 years old) are excluded from this survey.”

- Data Analysis Justification:

- Provide a rationale for selecting Mplus as the data analysis software and explain its suitability for testing the proposed hypotheses.

The reasons for using Mplus are explained in Section 3.1 of this article This paper collected data by distributing paper questionnaires offline. Because this study needs to investigate the mediating role of emotional exhaustion throughout the process and the moderating role of high commitment work system, while Mplus can directly implement the syntax instructions for the analysis of moderated effects of latent variables, it can obtain the regression coefficients and corresponding significance of the independent and dependent variables under low and high moderation variables, and can also draw pictures, which is more suitable for this study. Therefore, the data were analyzed and processed using SPSS 25.0 and Mplus statistical software to verify the hypotheses.

- Data Collection Method:

- Clarify Sampling Process: Detail how participants were recruited for the study (e.g., through an online survey or other method) to improve transparency.

A relevant statement is added in section 3.1 of the article: This paper collects data by distributing paper questionnaires offline

- Section 3.3: Questionnaire and Measurement Scales:

- Clarify "Mature Internal and External Articles": Explain what is meant by "mature internal and external articles." If this refers to validated sources or established scales, specify this for clarity.

Refer to the scale established and verified at home and abroad, see the second sentence of 3.3 Measurement of Variables in the article.

- Measurement Value Explanation: The phrase "The measured value of the variable was the average value of each item" needs further explanation. Specify whether this refers to scale scoring or another calculation.

Modify the expression as follows: "The measured value of the variable was measured by the average score of each item under the variable, this refers to scale scoring.”

- Clearly explain what "After Z-HCWS is removed" refers to in the table (1) to avoid confusion.

After discussion by the research team, the index "After Z-HCWS is removed" is not very relevant to the research content, so the expression of the column "after Z-HCWS" was deleted.

- The current interpretation lacks discussion of the findings with relating to exist research. Explain the significance of these results in the study’s broader context.

In the discussion of the results, add a discussion of the existing relevant findings, as described in section 5.1 of the paper.

- The in-text citations have not been correctly inserted. Please follow the appropriate reference style.

According to the publication requirements of Plos one, the full-text citation format was modified.

Reviewer #4: First of all, I would like to thank the authors for their efforts as well as for the opportunity to review the manuscript, "Research on the Influence Mechanism of Work-Family Conflict on Employees' Proactive Behaviors—A Moderated Mediator Model."

- The abstract is well-written and effectively represents the work that has been conducted.

- The research context is clearly identifiable.

- The theoretical elements are well introduced with appropriate bibliographic sourcing.

- The methodology used is relevant.

- The measurement tools employed are suitable, and the psychometric metrics are satisfactory.

- The discussion elements are well-structured and written, allowing for practical insights to be drawn from the results obtained.

1/ In the introduction, there are some instances of typos, for example, “job characteristics[3]between employees and organizations[4]few scholars have also[5].”

According to the modification suggestions, the words and grammar are modified as follows: “In exploring the relevant influencing factors of employee proactive behavior, most scholars focus on aspects within the work domain, such as leadership styles, job characteristics, the degree of alignment between employees and organizations [4-6].” See page 2 of the article.

2/ Generally, it seems that different citation methods have been used throughout the manuscript.

According to the publication requirements of plos one, the full-text citation format was modified.

3/ The manuscript would benefit from simplifying certain introductory elements, particularly section 2.2, “The mediating effect of emotional exhaustion on employees' proactive behavior,” whose extensiveness may lead to confusion. Therefore, I kindly ask you to consider reducing the density of this section for clarity.

The content of 2.2 is deleted and restated as described in 2.2

4/ In the methods section, the manuscript would benefit from standardizing the way numerical data is reported (e.g., “26.92

---

## [Decision Letter · Decision Letter 1]

29 Dec 2024

PONE-D-24-42561R1The Influence Mechanism of Work-Family Conflict on Employees' Proactive Behaviors-A Moderated Mediator ModelPLOS ONE

Dear Dr. Wan,

Thank you for submitting your manuscript to PLOS ONE. After careful consideration, we feel that it has merit but does not fully meet PLOS ONE’s publication criteria as it currently stands. Therefore, we invite you to submit a revised version of the manuscript that addresses the points raised during the review process.

We look forward to receiving your revised manuscript.

Kind regards,

Jianpeng Fan

Academic Editor

PLOS ONE

Journal Requirements:

Reviewers' comments:

Reviewer's Responses to Questions

**Comments to the Author**

1. If the authors have adequately addressed your comments raised in a previous round of review and you feel that this manuscript is now acceptable for publication, you may indicate that here to bypass the “Comments to the Author” section, enter your conflict of interest statement in the “Confidential to Editor” section, and submit your "Accept" recommendation.

Reviewer #2: (No Response)

Reviewer #3: All comments have been addressed

2. Is the manuscript technically sound, and do the data support the conclusions?

Reviewer #2: Partly

Reviewer #3: Yes

3. Has the statistical analysis been performed appropriately and rigorously? 

Reviewer #2: Yes

Reviewer #3: Yes

4. Have the authors made all data underlying the findings in their manuscript fully available?

Reviewer #2: Yes

Reviewer #3: Yes

5. Is the manuscript presented in an intelligible fashion and written in standard English?

Reviewer #2: No

Reviewer #3: Yes

6. Review Comments to the Author

Reviewer #2: 1.Typing and language errors persist. Please go through the whole document carefully. Such as, after reference [19], word " eployees" is there.

2."Some of the items were inconvenient to be expressed in one item Control variables" is very ambigous. What does that mean?

3. Gpower and sample validation is still missing.

4.Standard error is not required for normality.

5.Provide IRB statement and letter of consent

Reviewer #3: Thank you for addressing most of my comments. I hope the revisions improve your manuscript. Below are some additional suggestions that may further enhance your work:

1. It is recommended to refer to studies as citations in "5.1 Main Conclusions and Contributions." For instance, in the statement, “Different from the previous studies on employee proactive behavior from the perspective of social…”, and in “Third, in the existing studies on the high-commitment work system…”, you should cite the relevant research to substantiate your arguments.

2. Please resolve instances of "Error! Reference source not found." in the manuscript to ensure all citations are correctly formatted.

3. Certain sentences require grammatical corrections for improved readability. For example:

- “In order to test hypothesis 3 and 4. In this study, Mplus software was used to analyze the data.”

- “Emotional exhaustion (EE) is the intermediary variable at the individual level, and employee active behavior (PB) is the dependent variable at the individual level for SEM path analysis, as shown in the following table ”

- “The reliability analysis results are shown in Table 1.” it should be Table 3.

4. You mentioned: (“Control variables according to the demographic characteristics variables proposed by previous studies that may be related to the subject of this study to some extent, this paper mainly selects employee Gender, Age, Education, and LNTenure as the control variables.”)

In this case, these variables should be analyzed as control variables in a manner consistent with other variables in your analysis.

5. you stated that :"this paper systematically reviews the literature,".. If your paper includes a systematic literature review, consider illustrating the review process using the PRISMA criteria for clarity and methodological rigor.

6. Throughout the manuscript, there are grammar, punctuation, and wording errors. It is highly recommended to have the manuscript reviewed and proofread by a native English language expert.

I wish you all the best with your manuscript and its publication.

7. PLOS authors have the option to publish the peer review history of their article (what does this mean? ). If published, this will include your full peer review and any attached files.

**Do you want your identity to be public for this peer review?** For information about this choice, including consent withdrawal, please see our Privacy Policy .

Reviewer #2: **Yes: ** yumna ali

Reviewer #3: **Yes: ** Badr Albaram

---

## [Author Response · Author response to Decision Letter 2]

14 Jan 2025

We would like to thank the experts for reviewing the article in their busy schedules. The research team has carefully checked the citations of the article and the literature cited in the article is complete with no retracted articles cited.

Reviewer #2:

1. Typing and language errors persist. Please go through the whole document carefully. Such as, after reference [19], word " eployees" is there.

The language throughout the text was carefully revised and checked to ensure that there were no grammatical or spelling errors.

2."Some of the items were inconvenient to be expressed in one item Control variables" is very ambigous. What does that mean?

This was a linguistic error, and the relevant content has been rephrased, as detailed in section 3.3 of the article.

3.G.power and sample validation is still missing.

Thank you very much to the experts for taking time out of their busy schedules to view my paper and now some explanations on the relevant content. The research in this paper uses SPSS and MPLUS software to analyze the data, plus software has no special requirements for sample data, and the article uses structural equation modeling to analyze the data, there is no need for sample size estimation. In addition, the sampling of data for this questionnaire survey meets the needs of the research, and the relevant information has been explained in the article. After intense discussion with the research team, there is no need to use G Power analysis or sample validation.

4. Standard error is not required for normality.

The reference to standard errors was deleted.

5.Provide IRB statement and letter of consent

Submitted IRB statement and letter of consent in the submission system, in the supporting document, titled: plos one ethics approval. and rephrased the relevant content in 3.1 Research Methodology of the article.

Reviewer #3: Thank you for addressing most of my comments. I hope the revisions improve your manuscript. Below are some additional suggestions that may further enhance your work:

1. It is recommended to refer to studies as citations in "5.1 Main Conclusions and Contributions." For instance, in the statement, “Different from the previous studies on employee proactive behavior from the perspective of social…”, and in “Third, in the existing studies on the high-commitment work system…”, you should cite the relevant research to substantiate your arguments.

Relevant references have been added to support these two references, see references 60 and 61 for details.

2. Please resolve instances of "Error! Reference source not found." in the manuscript to ensure all citations are correctly formatted.

Changes were made to the citation formatting throughout the text.

3. Certain sentences require grammatical corrections for improved readability. For example:

- “In order to test hypothesis 3 and 4. In this study, Mplus software was used to analyze the data.”

- “Emotional exhaustion (EE) is the intermediary variable at the individual level, and employee active behavior (PB) is the dependent variable at the individual level for SEM path analysis, as shown in the following table ”

- “The reliability analysis results are shown in Table 1.” it should be Table 3.

We thank the experts for carefully reading and reviewing the article in their busy schedules, and the research team members have confidently read and revised the presentation of the whole article to ensure the accuracy of the expression.

4. You mentioned: (“Control variables according to the demographic characteristics variables proposed by previous studies that may be related to the subject of this study to some extent, this paper mainly selects employee Gender, Age, Education, and LNTenure as the control variables.”)In this case, these variables should be analyzed as control variables in a manner consistent with other variables in your analysis.

Relevant analyses of control variables have been added, as detailed in section 4.5 of the text.

5. you stated that :"this paper systematically reviews the literature,".. If your paper includes a systematic literature review, consider illustrating the review process using the PRISMA criteria for clarity and methodological rigor.

The lack of a systematic literature review in this paper was an expression error, and the relevant formulation has been revised, as detailed in the first sentence of 3.3.

6. Throughout the manuscript, there are grammar, punctuation, and wording errors. It is highly recommended to have the manuscript reviewed and proofread by a native English language expert.

The full text has been revised to ensure that the manuscript matches the English context and expression.

---

## [Decision Letter · Decision Letter 2]

21 Jan 2025

PONE-D-24-42561R2The Influence Mechanism of Work-Family Conflict on Employees' Proactive Behaviors-A Moderated Mediator ModelPLOS ONE

Dear Dr. Wan,

Thank you for submitting your manuscript to PLOS ONE. After careful consideration, we feel that it has merit but does not fully meet PLOS ONE’s publication criteria as it currently stands. Therefore, we invite you to submit a revised version of the manuscript that addresses the points raised during the review process.

We look forward to receiving your revised manuscript.

Kind regards,

Jianpeng Fan

Academic Editor

PLOS ONE

Journal Requirements:

Reviewers' comments:

Reviewer's Responses to Questions

**Comments to the Author**

1. If the authors have adequately addressed your comments raised in a previous round of review and you feel that this manuscript is now acceptable for publication, you may indicate that here to bypass the “Comments to the Author” section, enter your conflict of interest statement in the “Confidential to Editor” section, and submit your "Accept" recommendation.

Reviewer #2: (No Response)

Reviewer #3: All comments have been addressed

2. Is the manuscript technically sound, and do the data support the conclusions?

Reviewer #2: Partly

Reviewer #3: (No Response)

3. Has the statistical analysis been performed appropriately and rigorously? 

Reviewer #2: No

Reviewer #3: I Don't Know

4. Have the authors made all data underlying the findings in their manuscript fully available?

Reviewer #2: Yes

Reviewer #3: Yes

5. Is the manuscript presented in an intelligible fashion and written in standard English?

Reviewer #2: No

Reviewer #3: No

6. Review Comments to the Author

Reviewer #2: Dear author,

Please add rationale for not mentioning the minimum sample size for structual equtation modelling with citation.

Kindly, draw on more detail for sampling. The inclusion criteria needs to be added for companies/enterprises and teams.

Please eradicate any casual pharases like " so on"

Reviewer #3: I strongly advise the authors to check the manuscript with a native proofreader as it has a lot mistakes. These are some:

Title: Consider using a colon (:) instead of a dash (–) before "A Moderated Mediator Model" to align with proper title formatting.

Keywords: Ensure consistency in capitalization.

Introduction: Rewrite the phrase "businesses-focused HRM tactics" to include the full term before introducing the abbreviation for HRM.

Literature Review: Include the year of publication in parentheses after mentioning the authors. For example:

• "According to empirical research, Xu Yan (XXXX) discovered that work-family conflict has a significant negative impact on employees' work performance."

• "Chinese scholar Li Yifei et al. (XXXX) discovered through empirical analysis that work-family conflict negatively affects job satisfaction."

• "Zhang Junwei et al. (XXXX), using the resource conservation theory, found that work-family conflict significantly affects performance."

• Similarly, for Min's research, add: "Min (XXXX) stated that a high-commitment work system encourages initiative and creativity while reducing negative emotions."

2.2 The Mediating Effect of Emotional Exhaustion on Employees' Proactive Behavior: Ensure the indentation format of this section is consistent with other subheadings in the manuscript.

Clarifications and Edits:

• Proofread several sentences such as: " A person who experiences negative emotional reactions will be less enthusiastic and have a worse attitude at work, which will lead to burnout and a loss of commitment to the company. could possibly trigger the trend toward turnover [34].

• Replace "data" with appropriate word in: "In light of this data, we put forth the following hypothesis."

• Improve punctuation and flow, for example in: " putting in place employee involvement plans and flexible work designs to encourage diverse communication and hearing what staff members have to say [40] Employees are viewed as partners in the high-commitment work system, and the company can truly understand their requirements and communicate the corporate goal to them through active communication and other channels. to be acknowledged by workers and, via emotional dedication, to achieve the shared objectives and values of the company and workers."

• Replace "theories" with "hypotheses" in: "In light of this, the following hypotheses are put forth."

• Rephrase the paragraphs to place the hypothesis statement at the end. For example, in the sentence: " Hypothesis 3: The impact of emotional weariness on proactive conduct by employees is negatively moderated by a high-commitment work structure. This means that companies that have adopted a high-commitment work system have less of an effect on employees' proactive behavior than companies that have not adopted one or that have low levels of commitment in their HRM practices."

Clarifications on Terminology:

Clarify whether you are referring to "theory" or "hypothesis" in all applicable sentences, for example: " In addition to the aforementioned theory, we propose that the degree of high-commitment work systems that firms have in place may also have an impact on how work-family conflict affects employees' active behavior through emotional weariness."

And proofread and rephrase this sentence for clarity and accuracy: "Establishing a high-commitment work system can provide employees with human capital (skills and knowledge needed for tasks), emotional resources (positive emotional states), and social resources (a sense of security, accomplishment, and self-realization), encouraging productive behaviors that benefit the organization."

And Clarify the term "LNTenure."

And The sign "—" (dash) appears frequently in the manuscript. Proofread and replace it with a consistent and appropriate symbol.

4.5 Hypothesis Testing: Use plural form (e.g., "Hypotheses Testing") for this section title.

Tables: Ensure all tables are properly aligned and formatted to fit within the page, such as Table 12 and Table 13.

Citations: When mentioning "studies" or "research," always include proper citations to support the statements.

Variables items: Please list all items or state sample for each variable for items that you used in your adapted/adopted items, for clarity and credibility

7. PLOS authors have the option to publish the peer review history of their article (what does this mean? ). If published, this will include your full peer review and any attached files.

**Do you want your identity to be public for this peer review?** For information about this choice, including consent withdrawal, please see our Privacy Policy .

Reviewer #2: **Yes: ** yumna ali

Reviewer #3: **Yes: ** Badr Albaram

---

## [Author Response · Author response to Decision Letter 3]

19 Feb 2025

A sample size analysis was added in this study and the results showed that the conformed to the study sample size was sufficient.

The language of the paper has been proofread.

---

## [Decision Letter · Decision Letter 3]

3 Mar 2025

The Influence Mechanism of Work-Family Conflict on Employees' Proactive Behaviors-A Moderated Mediator Model

PONE-D-24-42561R3

Dear Dr. Wan,

We’re pleased to inform you that your manuscript has been judged scientifically suitable for publication and will be formally accepted for publication once it meets all outstanding technical requirements.

Kind regards,

Jianpeng Fan

Academic Editor

PLOS ONE

Additional Editor Comments (optional):

Reviewers' comments:

Reviewer's Responses to Questions

**Comments to the Author**

1. If the authors have adequately addressed your comments raised in a previous round of review and you feel that this manuscript is now acceptable for publication, you may indicate that here to bypass the “Comments to the Author” section, enter your conflict of interest statement in the “Confidential to Editor” section, and submit your "Accept" recommendation.

Reviewer #3: All comments have been addressed

2. Is the manuscript technically sound, and do the data support the conclusions?

Reviewer #3: Yes

3. Has the statistical analysis been performed appropriately and rigorously? 

Reviewer #3: Yes

4. Have the authors made all data underlying the findings in their manuscript fully available?

Reviewer #3: Yes

5. Is the manuscript presented in an intelligible fashion and written in standard English?

Reviewer #3: Yes

6. Review Comments to the Author

Reviewer #3: All the best! The authors have addressed our comments. However, there are minor typos that I believe will be reviewed before publication.

7. PLOS authors have the option to publish the peer review history of their article (what does this mean? ). If published, this will include your full peer review and any attached files.

**Do you want your identity to be public for this peer review?** For information about this choice, including consent withdrawal, please see our Privacy Policy .

Reviewer #3: **Yes: ** Badr Albaram

---

## [Editor Report · Acceptance letter]

PONE-D-24-42561R3

PLOS ONE

Dear Dr. Wan,

I'm pleased to inform you that your manuscript has been deemed suitable for publication in PLOS ONE. Congratulations! Your manuscript is now being handed over to our production team.

Kind regards,

on behalf of

Dr. Jianpeng Fan

Academic Editor

PLOS ONE